# Deep Neural Networks with Box Convolutions

**Egor Burkov** [1,2]               **Victor Lempitsky** [1,2]

[1] Samsung AI Center
[2] Skolkovo Institute of Science and Technology (Skoltech)
Moscow, Russia

## Abstract

Box filters computed using integral images have been part of the computer vision toolset for a long time. Here, we show that a convolutional layer that computes box filter responses in a sliding manner can be used within deep architectures, whereas the dimensions and the offsets of the sliding boxes in such a layer can be learned as a part of an end-to-end loss minimization. Crucially, the training process can make the size of the boxes in such a layer arbitrarily large without incurring extra computational cost and without the need to increase the number of learnable parameters. Due to its ability to integrate information over large boxes, the new layer facilitates long-range propagation of information and leads to the efficient increase of the receptive fields of network units. By incorporating the new layer into existing architectures for semantic segmentation, we are able to achieve both the increase in segmentation accuracy as well as the decrease in the computational cost and the number of learnable parameters.

## 1   Introduction

High-accuracy visual recognition requires integrating information from spatially-distant locations in the visual field in order to discern and to analyze long-range correlations and regularities. Achieving such long-range integration inside convolutional networks (ConvNets), which lack feedback connections and rely on feedforward mechanisms, is challenging. Modern ConvNets therefore combine several ideas that facilitate spatial long-range integration and allow to boost the effective size of receptive fields of the convolutional units. These ideas include stacking a very large number of layers (so that the local information integration effects of individual convolutional layers are accumulated) as well as using spatial downsampling of the representations (implemented using pooling or strides) early in the pipeline.

One particularly useful and natural idea is the use of spatially-large filters inside convolutional layers. Generally, a naive increase of the spatial filter size leads to the quadratic increase in the number of learnable parameters and numeric operations, leading to architectures that are both slow and prone to overfitting. As a result, most current ConvNets rely on small $3 \times 3$ filters (or even smaller ones) for most convolutional layers [25]. A highly popular alternative to the naive enlargement of filters is dilated/"à trous" convolutions [13, 3, 33] that expand filter sizes without increasing the number of parameters by padding them with zeros. Many popular architectures, especially semantic segmentation architectures that emphasize efficiency, rely on dilated convolutions extensively (e.g. [3, 33, 34, 20, 21]).

Here, we present and evaluate a new simple approach for inserting convolutions with large (potentially very large) spatial filters inside ConvNets. The approach is based on box filtering and relies on integral images [18], as many classical works in computer vision do [31]. The new *convolutional box layer* applies box average filters in a convolutional manner by sliding the 2D axes-aligned boxes across spatial locations, while performing box averaging at every location. The dimensions and the

offsets of the boxes w.r.t. the sliding coordinate frame are treated as learnable parameters of this layer. The new layer therefore combines the following merits: (i) large-size convolutional filtering, (ii) low number of learnable parameters, (iii) computational efficiency achieved via integral images, (iv) effective integration of spatial information over large spatial extents.

We evaluate the new layer by embedding it into a block that includes the new layer, as well as a residual connection and a $1 \times 1$ convolution. We then consider semantic segmentation architectures that have been designed for optimal accuracy-efficiency trade-offs ( *E-Net* [20] and *ERF-Net* [21]), and replace analogous blocks based on dilated convolutions inside those architectures with the new block. We show that such replacement allows both to increase the accuracy of the network and to decrease the number of operations and the number of learnable parameters inside the networks. We conclude that the new layer (as well as the proposed embedding block) are viable solutions for achieving efficient spatial propagation and integration of information in ConvNets as well as for designing efficient and accurate ConvNets.

## 2 Related work

**Spatial propagation in ConvNets.** In order to increase receptive fields of the convolutional neurons, and to effectively propagate/mix information across spatial locations, several high-level ideas can be implemented in ConvNets (potentially, in combination). First, a ConvNet can be made very deep, so that the limited spatial propagation of individual layers is accumulated. E.g. most top-performing ConvNets have several dozens of convolutional layers [29, 12, 30]. Such "extreme" depth, however, comes at a price of high computational demands, which are at odds with lots of application domains, such as computer vision on low-power devices, autonomous driving etc.

The second idea that is invariably used in all modern architectures is downsampling, which can be implemented via pooling layers [17] or simply by strided convolutions [16, 27]. Spatial shrinking of the representations naturally makes integration of information from different parts of the visual field easier. At the same time, excessive downsampling leads to the loss of spatial information. This can be particularly problematic for such applications as semantic segmentation, where each downsampling has to be complemented by upsampling layers in order to achieve pixel-level predictions. Spatial information cannot usually be recovered very well by downsampling-upsampling (hourglass) architectures, and the addition of skip connections [19, 22] provides only partial remedy to this problem [33].

As discussed above, dilated convolutions (also known as à trous convolutions) [33, 3] are used extensively in order to expand the effective filter sizes and to speed-up propagation of spatial information during the inference in ConvNets. Another potential approach is to introduce non-local non-convolutional layers that can be regarded as the incorporation of Conditional Random Fields (CRF) inference steps into the network. These include layers that emulate mean field message propagation [35, 4] as well as Gaussian CRF layers [2].

Our approach introduces yet another approach for spatial propagation/integration of information in ConvNets that can be combined with the approaches outlined above. Our comparison with dilated convolutions suggests that the new approach is competitive and may be valuable for the design of efficient ConvNets.

**Box Filters in Computer Vision.** Our approach is based on the box filtering idea that has long been mainstream in computer vision. Through 2000s, a large number of architectures that use box filtering to integrate spatial context information have been proposed. The landmark work that started the trend was the Viola-Jones face detection system [31]. Later, this idea was extended to pedestrian detectors [32]. Two-layer architectures that applied box filtering on top of other transforms such as texton filters or edge detectors became popular by the end of that decade [24, 9]. Box-filtered features remain a popular choice for building decision-tree based architectures [23, 8]. All these (and hundreds of other works) capitalized on the ability of box filtering to be performed very efficiently through the use of the integral image trick [18].

Given the success of integral-image based box filtering in the "pre-deep learning era", it is perhaps surprising that very few attempts have been made to insert such filtering into ConvNets. Various methods that perform sum/average pooling over large spatial boxes have been proposed for deep object detection [11], semantic segmentation [34], and image retrieval [1]. All those systems,

however, apply box filtering only at one point of the pipeline (typically towards the very end after the convolutional part), do so in a non-convolutional manner, and do not rely on the integral image trick since the number of boxes over which the summation is performed is usually limited. Integral image-based filtering has been applied to pool of deep features over sliding windows in [10]. Rather differently to our method, [10] use fixed predefined sizes of the boxes, and use average box filtering only as a penultimate layer in their network, which predicts the objectness score. In contrast, our approach learns the coordinates of the boxes in an end-to-end manner, and provides a generic layer that can be inserted into a ConvNet architecture multiple times. Our experiments verify that learning box coordinates is important for achieving good performance.

## 3  Method

This section describes the Box Convolution layer and discusses its usage in ConvNets in detail. Note that while the idea behind the new layer is rather simple (implement box averaging in a convolutional manner; use integral images to speed up convolutions), an important part of our approach is to make the coordinates of the boxes learnable. This requires us to consider continuous-valued box coordinates, unlike the approaches in classic computer vision, which invariably consider integer-valued box coordinates.

### 3.1  Box Convolution Layer

We start by defining the *box averaging kernel* with parameters $\theta = (x_{\min}, \ x_{\max}, \ y_{\min}, \ y_{\max})$ as the following function over 2D plane $\mathbb{R}^2$:

$$K_\theta\left(x,y\right) = \frac{\mathbb{I}\left(x_{\min} \leq x \leq x_{\max}\right)\mathbb{I}\left(y_{\min} \leq y \leq y_{\max}\right)}{\left(x_{\max} - x_{\min}\right)\left(y_{\max} - y_{\min}\right)} \tag{1}$$

Here, $x_{\min} < x_{\max}$, $y_{\min} < y_{\max}$ are the dimensions of the box averaging kernel (jointly denoted as $\theta$), and $\mathbb{I}$ is the indicator function. The kernel naturally integrates to one, and convolving a function with such kernel corresponds to a low-pass averaging transform.

**Forward Pass.**  The box convolution layer takes as an input the set of $N$ convolutional maps (2D tensors) and applies $M$ different box kernels to each of the $N$ incoming channels, thus resulting in $NM$ output convolutional maps (2D tensors). The layer therefore has has $4NM$ learnable parameters $\{\theta_n^m\}_{n=1,m=1}^{N,M}$.

The input maps are defined over a discrete lattice $\left(\hat{\mathbf{I}}_{i,j}\right)_{i=1,j=1}^{w,h}$, where $h$ and $w$ are image dimensions. In order to apply box averaging corresponding to continuous $\theta$ in (1) we extend each of the input maps to the continuous plane. We use a piecewise-constant approximation and zero padding for this extension as follows:

$$\mathbf{I}\left(x,y\right) = \left\{ \begin{array}{ll} \hat{\mathbf{I}}_{[x],[y]}, & 1 \leq [x] \leq h \text{ and } 1 \leq [y] \leq w; \\ 0, & \text{otherwise} \end{array} \right. , \tag{2}$$

where $[\cdot]$ denotes rounding to the nearest integer. Here $\hat{\mathbf{I}}$ denotes one of the input channels and $\mathbf{I}$ denotes its extension onto the plane.

Let $\hat{\mathbf{O}}$ be one of the output channels corresponding to the input channel $\hat{\mathbf{I}}$ and let $\theta = (x_{\min}, \ x_{\max}, \ y_{\min}, \ y_{\max})$ be the corresponding box coordinates. To find $\hat{\mathbf{O}}$ we naturally apply the convolution (correlation) with the kernel (1), and then convert the output back to the discrete representation by sampling the result of the convolution at the lattice vertices. Overall, this corresponds to the following transformation:

$$\hat{\mathbf{O}}_{x,y} = \mathbf{O}\left(x,y\right) = \int\limits_{-\infty}^{+\infty}\int\limits_{-\infty}^{+\infty} \mathbf{I}(x+u,y+v)K_\theta(u,v)\,\mathrm{d}u\,\mathrm{d}v = \tag{3}$$

$$= \frac{1}{\left(x_{\max} - x_{\min}\right)\left(y_{\max} - y_{\min}\right)} \int\limits_{x+x_{\min}}^{x+x_{\max}}\int\limits_{y+y_{\min}}^{y+y_{\max}} \mathbf{I}(u,v)\,\mathrm{d}u\,\mathrm{d}v\,, \tag{4}$$

where $\mathbf{O}$ denotes the continuous result of the convolution.

Note that while our construction here uses zero-padding in (2), more sophisticated padding schemes are possible. Our experiments with such schemes, however, did not result in better performance.

**Backpropagation to the inputs.** Since the transformation described above is a convolution (correlation), the gradient of some loss $L$ (e.g. a semantic segmentation loss) w.r.t. its input can be obtained by computing the correlation of loss gradients w.r.t. its output with flipped kernels $K_\theta(-x, -y)$, where the contributions from the $N$ output channels corresponding to the same input channels are accumulated additively:

$$\frac{\partial L}{\partial \hat{\mathbf{I}}_{x,y}} = \sum_{n=1}^{N} \int_{-\infty}^{+\infty} \int_{-\infty}^{+\infty} \mathbf{G}^n(x+u, y+v) K_{\theta_n}(-u, -v) \, \mathrm{d}u \, \mathrm{d}v \,, \tag{5}$$

where $\theta_1, \ldots, \theta_N$ are the layer's parameters used to produce output channels $\hat{\mathbf{O}}^1, \ldots, \hat{\mathbf{O}}^N$ respectively, and $\mathbf{G}^n(x, y)$ is the continuous domain extension of $\frac{\partial L}{\partial \hat{\mathbf{O}}^n}$ as in (2).

**Backpropagation to the parameters $\theta$.** The expression for the partial derivative $\frac{\partial L}{\partial x_{\max}}$ is derived in the Appendix and evaluates to:

$$\frac{\partial L}{\partial x_{\max}} = -\frac{1}{(x_{\max} - x_{\min})} \sum_{x=1}^{h} \sum_{y=1}^{w} \frac{\partial L}{\partial \hat{\mathbf{O}}_{x,y}} \hat{\mathbf{O}}_{x,y} + \tag{6}$$

$$+ \frac{1}{(x_{\max} - x_{\min})(y_{\max} - y_{\min})} \sum_{x=1}^{h} \sum_{y=1}^{w} \frac{\partial L}{\partial \hat{\mathbf{O}}_{x,y}} \int_{y+y_{\min}}^{y+y_{\max}} \mathbf{I}(x+x_{\max}, v) \, \mathrm{d}v.$$

Partial derivatives for $x_{\min}, y_{\min}, y_{\max}$ have analogous expressions.

**Initialization and regularization.** Unlike many common ConvNet layers, ours does not allow arbitrary real parameters $\theta$. Thus, during optimization we ensure positive widths and heights. During learning, we ensure that the width and the height are at least $\epsilon = 1$ pixel: $x_{\max} - x_{\min} > \epsilon$, $y_{\max} - y_{\min} > \epsilon$. These constraints are ensured in a projective gradient descent fashion. E.g. when the first of these constraints gets violated, we change $x_{\min}$ and $x_{\max}$ by moving them away from each other to the distance $\epsilon$, while preserving the midpoint. We also clip the coordinates, when they go outside the $[-w; +w]$ range (for $x_{\min}$ and $x_{\max}$) or $[-h; +h]$ range (for $y_{\min}$ and $y_{\max}$).

Additionally, we impose a standard L2-regularization $\frac{\lambda}{2} \|\theta\|^2$ on the parameters of all box layers, thus shrinking the box dimensions towards zero at each step. Among other things, such regularization prevents both instabilities (often associated with some filters growing very large) as well as the emergence of degenerate solutions, where boxes drift completely outside of the image boundaries for all locations.

To initialize $\theta$, we aim to diversify initial filters, so we initialize them randomly and independently. We first sample a center point of a box uniformly from the rectangle $\mathcal{B} = \left[-\frac{w}{2}; \frac{w}{2}\right] \times \left[-\frac{h}{2}; \frac{h}{2}\right]$ and then uniformly sample the width and the height so that $[x_{\min}; x_{\max}] \subset \left[-\frac{w}{2}; \frac{w}{2}\right]$ and $[y_{\min}; y_{\max}] \subset \left[-\frac{h}{2}; \frac{h}{2}\right]$. Such choice of initial parameters ensures enough potentially non-zero output pixels in the resulting layer (which leads to strong enough gradient during learning).

**Fast computation via integral images.** The forward-pass computation (4) as well as the backward-pass computations (5) and (6) involve integration over 1D axes aligned intervals and 2D axes aligned boxes. To enable fast computation, we split each interval of the integration into the integral part and the two fractional parts in the end. As a result all integrals can be approximated using box sums and line sums. For example, if $x_{\max} - x_{\min} > 1$, then the integral in (4) can be estimated as a box sum $\sum_{i=\lceil x+x_{\min}\rceil}^{\lfloor x+x_{\max}\rfloor} \sum_{j=\lceil y+y_{\min}\rceil}^{\lfloor y+y_{\max}\rfloor} \hat{\mathbf{I}}(i, j)$ plus a weighted sum of four line sums, e.g. $\left(\lceil x+x_{\max}\rceil + \frac{1}{2} - x - x_{\max}\right) \sum_{j=\lceil y+y_{\min}\rceil}^{\lfloor y+y_{\max}\rfloor} \hat{\mathbf{I}}_{\lceil x+x_{\min}\rceil, j}$,

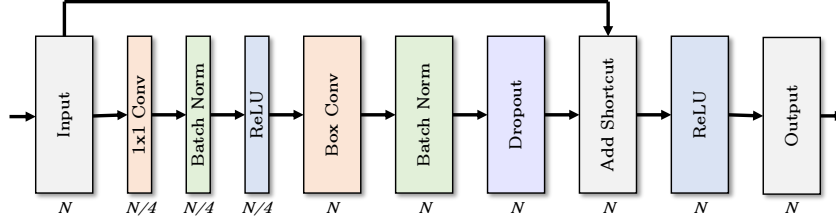

Figure 1: The block architecture that is used to embed box convolution into our architectures. For the sake of simplicity, the specific narrowing factor of 4 is used. The block combines $1 \times 1$ convolution that shrinks the number of channels from $N$ to $N/4$, and then box convolution that increases the number of channels from $N/4$ to $N$.

plus four scalars corresponding to the corners of the domain, e.g. $\left([x + x_{\max}] + \frac{1}{2} - x - x_{\max}\right)\left([y + y_{\max}] + \frac{1}{2} - y - y_{\max}\right) \hat{\mathbf{I}}_{[x+x_{\max}],[y+y_{\max}]}$. Box sums and line sums can then be handled efficiently using the integral image $\mathbf{I}_{\int}(x, y) = \sum_{i \leq x} \sum_{j \leq y} \hat{\mathbf{I}}_{x,y}$. The backprop step is handled analogously, as $\mathbf{I}_{\int}$ is reused to compute (6), and the integral image for $\frac{\partial L}{\partial \tilde{\mathbf{O}}_{x,y}}$ is computed to evaluate the integrals in (5).

Integral image computation on GPU is performed by first performing parallel cumulative sum computation over columns, then transposing the result and accumulating over the new columns (former rows) again, and finally transposing back.

**Embedding box convolutions into an architecture.** The derived box convolution layer acts on each of the input channels independently. It is therefore natural to interleave box convolutions with cross-channel $1 \times 1$ convolution, making the resulting block in many ways similar to the depthwise separable convolution block [5]. We further expand the block by inserting the standard components, namely ReLU non-linearities, batch normalizations [14], a dropout layer [28] and a residual connection [12]. The resulting block architecture is shown in Figure 1. The block transforms an input stack of $N$ channels into the same-size stack of $N$ channels. Notably, the majority of the learnable parameters (and the majority of the floating-point operations) are in the $1 \times 1$ convolution, which has $O(N^2)$ complexity per each spatial position. All remaining layers including the box convolution layer, which acts within each channel independently, have $O(N)$ complexity per spatial location.

## 4 Experiments

We evaluate and analyze the performance of the new layer for the task of semantic segmentation, which is among vision tasks that are most sensitive to the ability to propagate contextual evidence and to preserve spatial information. Also, convolutional architectures for semantic segmentation have been studied extensively over the last several years, resulting in strong baseline architectures.

**Base architectures and datasets.** To perform the evaluation of the new layer and the embedding block, we consider two base architectures for semantic segmentation: **ENet** [20] and **ERFNet** [21]. Both architectures have been designed to be accurate and at the same time very efficient. They both consist of similar residual blocks and feature dilated convolutions. In our evaluation, we replace several of such blocks with the new block (Figure 1).

Both ENet and ERFNet have been fine-tuned to perform well on the Cityscapes dataset for autonomous driving [7]. The dataset ("fine" version) consists of 2975 training, 500 validation and 1525 test images of urban environments, manually annotated with 19 classes. This dataset represents one of the main benchmarks for semantic segmentation, with a very large number of evaluated architectures. The ENet architecture has also been tuned for the SUN RGB-D dataset [26], which is a popular benchmark for indoors semantic segmentation and consists of 5050 train and 5285 test images, providing annotation for 37 object and stuff classes. Following the original paper [20], we train all architectures on RGB data only ignoring depth.

| | Validation set | | | | | Test set | | |
|---|---|---|---|---|---|---|---|---|
| | **ENet** | **BoxENet** | **BoxENet†** | **ENet⁻** | **BoxOnlyENet** | **ENet** | **BoxENet** | **BoxOnlyENet** |
| **IoU-class, %** | 59.4 | **64.6** | 60.3 | 54.1 | 61.0 | 58.3 | **64.7** | 61.8 |
| **IoU-categ., %** | 81.8 | **83.2** | 80.9 | 80.9 | 81.8 | 80.4 | **83.8** | 82.1 |

| | Validation set | | | | Test set | |
|---|---|---|---|---|---|---|
| | **ERFNet** | **BoxERFNet** | **BoxERFNet†** | **ERFNet⁻** | **ERFNet** | **BoxERFNet** |
| **IoU-class, %** | 68.8 | **69.0** | 63.6 | 59.8 | 68.0 | **68.1** |
| **IoU-categ., %** | 85.3 | **85.4** | 84.1 | 84.1 | **86.5** | 85.6 |

Table 1: Results for ENet-based models (top) and ERFNet-based models (bottom) on the Cityscapes dataset. For ENet configurations, BoxENet considerably outperforms ENet as well as the ablations. For ERFNet, the version with boxes performs on par, while requiring less resources (Table 3). See text for more discussion.

| | **ENet** | **BoxENet** | **BoxENet†** | **ENet⁻** | **ERFNet** | **BoxERFNet** |
|---|---|---|---|---|---|---|
| **mean IoU** | 22.9% | **24.5%** | 21.9% | 13.2% | 25.3% | **28.7%** |
| **Class accuracy** | 34.0% | **37.2%** | 32.2% | 19.0% | 36.1% | **41.9%** |
| **Pixel accuracy** | 66.5% | **67.1%** | 64.2% | 59.7% | 68.6% | **69.0%** |

Table 2: Performance on the test set of the SUN RGB-D dataset (all architectures disregarded the depth channel). The networks with box convolutions perform better than the base architectures and than the ablations. In the case of ERFNet family, BoxERFNet also requires less resources than ERFNet (Table 3). See text for more discussion.

**New architectures.** We design two new architectures, namely *BoxENet* based on ENet and *Box-ERFNet* based on ERFNet. Both base architectures contain downsampling (based on strided convolution) and upsampling (based on upconvolution) layer groups. Between them, a sequence of residual blocks is inserted. For instance, ENet authors rely on bottleneck residual blocks, sometimes employing dilation in its $3 \times 3$ convolution, or replacing it by two 1D convolutions (also possibly dilated). It has four of these after the second downsampling group, 16 after the third group, two after the first upsampling and one after the second upsampling.

When designing BoxENet, we replace **every second block** in all these sequences by our proposed block, additionally replacing the very last block as well. As a matter of fact, in this way we replace all blocks with dilated convolutions, so that BoxENet does not have any dilated convolutions at all. The comparison between ENet and BoxENet thus effectively pits dilated convolutions versus box convolutions. We have further evaluated the configuration where all resolution-preserving blocks are replaced by our proposed block (*BoxOnlyENet*). We use the same bottleneck narrowing factor 4 (same as in the original ENet) as in the original blocks except for the very last block, where we squeeze $N$ channels to $N/2$. The dropout rate is set to 0.25 where the feature map resolution is lowest (1/8th of the input), and to 0.15 elsewhere.

ERFNet has a similar design to ENet, although the blocks in it operate at the full number of channels. Here, we implemented the above pattern with the following changes: (i) in the first resolution-preserving block sequence, only one original block is kept, (ii) from the last two sequences, one of the two blocks is simply removed without replacing it by our block. All our blocks have a narrowing factor of 4 (keeping this from the ENet experiments). This time we always use the exact same (possibly zero) dropout rate as in the corresponding original replaced block. In addition, we remove dilation from all the remaining blocks.

**Ablations.** The key aspect of the new box convolution layer is the learnability of the box coordinates. To assess the importance of this aspect, we have evaluated the ablated architectures *BoxENet†* and *BoxERFNet†* that are identical to BoxENet and BoxERFNet, but have the coordinates of the boxes in the box convolution layers frozen at initialization. Another pair of ablations, *ENet⁻* and *ERFNet⁻* are the modification that have the corresponding residual blocks removed rather than replaced with the new block (these ablations were added to make sure that the increase in accuracy were not coming simply through the reduction of learnable parameters).

**Performance.** We have used standard learning practices (ADAM optimizer [15], step learning rate policy with the same learning rate as in the original papers [20, 21]) and standard error measures.

|  | ENet | BoxENet | ENet$^-$ | BoxOnlyENet | ERFNet | BoxERFNet | ERFNet$^-$ |
|---|---|---|---|---|---|---|---|
| **MultAdds, billions** | 4.601 | 3.468 | 2.950 | 2.290 | 30.876 | 16.042 | 15.827 |
| **CPU time, ms** | 558 | 478 | 338 | 414 | 1682 | 822 | 772 |
| **GPU time, ms** | 32.8 | 33.1 | 21.2 | 33.6 | 78.4 | 44.0 | 39.7 |
| **# of params, millions** | 0.356 | 0.243 | 0.201 | 0.124 | 2.059 | 1.040 | 1.020 |

Table 3: Resource costs for the architectures. Timings are averaged over 40 runs. See text for discussion.

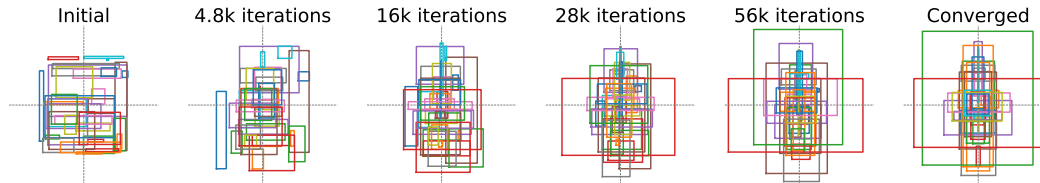

Figure 2: Change of the box coordinates of the first box convolution layer in BoxENet as the training on Cityscapes progresses. The boxes with the biggest importance (at the end of training) are shown.

Once box-based architectures are trained, we round the box coordinates to the nearest integers, fix them and fine-tune the network, so that at test time we do not have to deal with fractional parts of integration intervals. Such rounding post-processing resulted in slightly lower runtimes (5.7% and 1.3% speedup for BoxENet and BoxERFNet respectively) with essentially same performance. Our approach was implemented using Torch7 library [6].

The comparison on the validation set of the Cityscapes dataset are given in Table 1. We also report the performance of non-ablated variants on the test set. Table 2 further reports the comparison on the test set of the SUN RGB-D dataset (note that ENet accuracy on SUN RGB-D is higher than reported in the original paper due to the higher input image resolution). The new architectures (BoxENet/BoxERFNet) outperform the counterparts (ENet/ERFNet) considerably for the ENet family on both datasets and for the ERFNet family on the SUN-RGBD dataset. On the Cityscapes dataset, BoxERFNet achieves similar accuracy to ERFNet, however it has strong advantage in terms of computational resources (see below). BoxOnlyENet performs in-between ENet and BoxENet, suggesting that the optimal approach might be to combine standard 3x3 convolutions with box convolutions (as is done in BoxENet). Finally, fixing the bounding box coordinates to random initializations (BoxENet$^\dagger$ and BoxERFNet$^\dagger$) lead to significant degradation in accuracy, suggesting that the learnability of box coordinates is crucial.

We further compare the number of operations, the GPU and CPU inference times on a laptop (an NVIDIA GTX 1050 GPU with cuDNN 7.2, a single core of Intel i7-7700HQ CPU), and the number of learnable parameters in Table 3. The new architectures are more efficient in terms of the number of operations, and also in terms of GPU timings for the ERFNet case. The GPU computation time is marginally bigger for BoxENet compared to ENet despite BoxENet incurring fewer operations, as the level of optimization of our code does not quite reach the level of cuDNN kernels. On CPU, the new architectures are considerably faster (by a factor of two in the ERFNet case). Finally, the box-based architectures have considerably fewer learnable parameters compared to the base architecture.

**Box statistics.** To analyze the learning of box coordinates, we introduce the measure of box importance. For each box filter, its importance is defined as the average absolute weight of the corresponding channel in the subsequent $1 \times 1$ convolution multiplied by the maximum absolute weight corresponding to the input channel in the preceding convolution.

The evolution of boxes during learning is shown in Figure 2. We observe the following trends. Firstly, under the imposed regularization, a certain subset of boxes shrinks to minimal width and height. Detecting such boxes and eliminating them from the network is then possible at this point. While this should lead to improved computational efficiency, we do not pursue this in our experiments. Another obvious trend is the emergence of boxes that are symmetric w.r.t. the vertical axis. We observe this phenomenon to be persistent across layers, architectures and datasets. The effect persists even when horizontal flip augmentations are switched off during training. All this probably suggesting that a three-DOF parameterization $\theta = \{y_{\min}, y_{\max}, \text{width}\}$ can be used in place of the four-DOF parameterization.

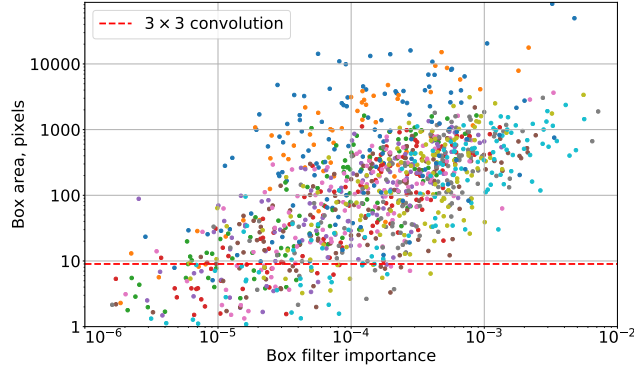

Figure 3: The vertical axis shows the areas of box filters learned on Cityscapes by the BoxENet network (note the log-scale). Colors depict different layers. The learned network contains very large (>10000 pixels) boxes and lots of boxes spanning many hundreds of pixels. Using filters of this size in a conventional ConvNet would be impractical from the computational and statistical viewpoints.

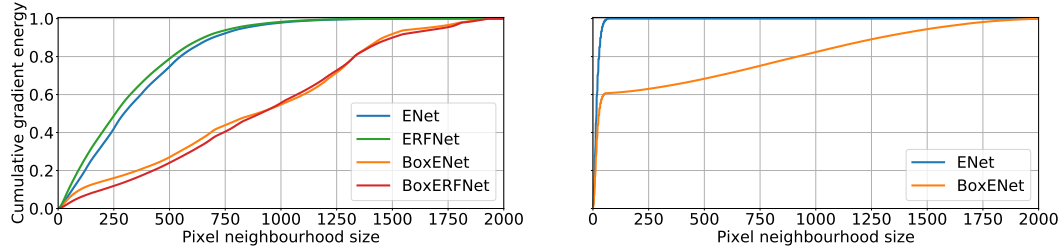

Figure 4: The curves visualizing the effective receptive fields of the output units of different architectures (left) and the shallow sub-networks of the BoxENet and ENet architectures, specifically first 7 (i.e. up to the first box convolution block) blocks of BoxENet and their ENet counterpart with the original 7$^{th}$ block (right). The models have been trained on the Cityscapes dataset. The effective receptive field can be thought as the radius (horizontal position) at which the curve saturates to 1 (see text for the details of the curve computation). Architectures with box convolutions have much bigger effective receptive fields.

The final state of the BoxENet trained on the Cityscapes dataset is visualized in Figure 3. The scatterplot shows the sizes (areas) of boxes in different layers plotted against the importances determined by the the activation strength of the input map and the connection strength to the subsequent layers. The network has converged to a state with a lot of large and very large box filters, and most of the very large filters are important for the subsequent processing. We conclude that the resulting network relies heavily on box averaging over large and very large extents. A standard ConvNet with similarly-sized filters of a general form would not be practical, as it would be too slow and the number of parameters would be too high.

**Receptive fields analysis.** We have analyzed the impact of new layers onto the effective size of the receptive fields. For this reason we have used the following measure. Given a spatial position $(p, q)$ inside the visual field, and a layer $k$, we consider the *gradient response map* $\hat{\mathbf{M}}(p, q, k, \hat{\mathbf{I}}^0) = \{\frac{1}{N_k} \sum_{i=1}^{N_k} \|\partial y_i(p, q)/\partial \hat{\mathbf{I}}_{x,y}^0\|\}_{x,y}$, where $y_i(p, q)$ is the unit at level $k$ of the network, corresponding to the $i$-th map at position $(p, q)$, $\hat{\mathbf{I}}^0$ is the input image, $(x, y)$ is the position in the input image.

The map $\hat{\mathbf{M}}(p, q, k, \hat{\mathbf{I}}^0)$ estimates the influence of various positions in the input image on the activation of the unit in the $k$-th channel at position $(p, q)$. The bigger the effective receptive fields of the units in layer $k$ in the network, the more spread-out will be the map $\hat{\mathbf{M}}(p, q, k, \hat{\mathbf{I}}^0)$ around positions $(p, q)$. We measure the spread by taking a random image from the Cityscapes dataset, considering a random location $(p, q)$ in the visual fields, and computing the gradient response map $\hat{\mathbf{M}}(p, q, k, \hat{\mathbf{I}}^0)$ for a certain level $k$ of the network. For each computed map, we then consider a family of square windows of varying radius $r$ centered at $(p, q)$, integrate the map value over each window, and divide the integral over the total integral of the gradient response map, obtaining the value $E(r)$ between 0

and 1 (we refer to this value as *cumulative gradient energy*). One can think of the value $r$ at which $E(r)$ comes close to 1 as the radius of an effective receptive field (since the corresponding window contains nearly all gradients).

We then consider the curves $E(r)$, average them over 30 images and 20 random positions $(p, q)$. The curves for the final layer (prior to softmax) and an early layer (after the first seven convolutions) are shown in Figure 4. For networks with box convolutions, the cumulative curves saturate to 1 at much higher radii $r$ than for the networks relying on dilated convolutions (effectively for some locations and some images the effective receptive field spans the whole image). Overall, we conclude that networks with box convolutions have much bigger effective receptive fields, both for units in early layers as well as for the output units.

## 5 Summary

We have introduced a new convolutional layer that computes box filter responses at every location, while optimizing the parameters (coordinates) of the boxes within the end-to-end learning process. The new layer therefore combines the ability to aggregate information over large areas, the low number of learnable parameters, and the computational efficiency achieved via the integral image trick. We have shown that the learning process indeed leads to large boxes within the new layer, and that the incorporation of the new layer increases the receptive fields of the units in the middle of semantic segmentation networks very considerably, explaining the improved segmentation accuracy. What is more, this increase in accuracy comes alongside the reduction of the number of operations. The code of the new layer, as well as the implementation of BoxENet and BoxERFNet architectures, are available at the project website (`https://github.com/shrubb/box-convolutions`).

## Appendix

Here, we derive the backpropagation equation for $\frac{\partial L}{\partial x_{\max}}$. We start with the chain rule that suggests:

$$\frac{\partial L}{\partial x_{\max}} = \sum_{x=1}^{h} \sum_{y=1}^{w} \frac{\partial L}{\partial \hat{\mathbf{O}}_{x,y}} \cdot \frac{\partial \hat{\mathbf{O}}_{x,y}}{\partial x_{\max}}. \tag{7}$$

To compute the derivative of $\hat{\mathbf{O}}_{x,y}$ over $x_{\max}$, we use the product rule treating (4) as a product of a ratio coefficient and a double integral. The derivative of the former evaluates to:

$$\partial \left[ \frac{1}{(x_{\max} - x_{\min})(y_{\max} - y_{\min})} \right] / \partial x_{\max} = -\frac{1}{(x_{\max} - x_{\min})^2 (y_{\max} - y_{\min})} . \tag{8}$$

The double integral in (4) has $x_{\max}$ as one of its limits, and its derivative therefore evaluates to:

$$\partial \left[ \int_{x+x_{\min}}^{x+x_{\max}} \int_{y+y_{\min}}^{y+y_{\max}} \mathbf{I}(u, v) \, \mathrm{d}u \, \mathrm{d}v \right] / \partial x_{\max} = \int_{y+y_{\min}}^{y+y_{\max}} \mathbf{I}(x + x_{\max}, v) \, \mathrm{d}v. \tag{9}$$

Plugging both (8) and (9) into the product rule for expression (4), we get:

$$\frac{\partial \hat{\mathbf{O}}_{x,y}}{\partial x_{\max}} = \frac{1}{(x_{\max} - x_{\min})} \left( -\hat{\mathbf{O}}_{x,y} + \frac{1}{(y_{\max} - y_{\min})} \int_{y+y_{\min}}^{y+y_{\max}} \mathbf{I}(x + x_{\max}, v) \, \mathrm{d}v \right) \tag{10}$$

which allows us to expand (7) as (6).

**Acknowledgement.** Most of the work was done when both authors were full-time with Skolkovo Institute of Science and Technology. The work was supported by the Ministry of Science of Russian Federation grant 14.756.31.0001.

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
