[Reviews · NeurIPS 2018]

Reviewer 1



Thank you for the clarifications. I am inclined to keep my ratings. One concern I still have is the new approach (BoxERFNet) doesn't seem to be giving any gains in Cityscapes (test) so I not sure how effective the approach is. The results on the SUN RGBD are good but as mentioned, the earlier models were not tuned for that dataset, so it is hard to make a strong conclusion. ----------------------------------- The paper introduces a way to use integral images in neural nets. Integral images used a trick for fast computation and has been used for many problems in computer vision but had not been used in neural networks yet. The learning allows the convolution filter dimensions and positions to be learned and that too in a faster manner. The paper is easy to read and provides sufficient background to understand the problem and solution. The strength of the paper seems to be that using this technique can possibly reduce the computation time (but see concerns below) even though the segmentation accuracy does not increase in all instances. It would be useful if the authors address some of these concerns. It would be beneficial to summarize the contribution of the paper in a few bullet points. Line 131, define L. Why does Table 1 not include test results for ERFNet and BoxERFNet and Table 2 not contain ERFNet type results? The results of ERFNet and BoxERFNet seem to be very similar and so the integral images seem to only be beneficial in terms of reducing time. Are the runs in Tables especially Table 3 averaged over multiple runs. Because it is interesting to see that the integral images helps in the ERFNet but not ENet in terms of GPU time (or what does it mean that the optimization does not reach the level of cuDNN)? Do we get similar box positions and sizes during multiple runs on convergence. If not, are one set of results better than another? minor: line 271: The bigger the effective ... line 289: We have introduce"d" a new ... The supplementary material have some equation numbers missing (i.e. ??).

Reviewer 2



This paper introduced a new convolutional layer that computes box filter responses at every location. The papers shows that it is possible to optimise the parameters of the boxes (coordinates) within end-to-end learning process. Experiments show very marginal improvement over standard convolutional networks. I lead to accept this paper as a poster as I believe the way to optimise for the coordinates of box filters can be useful for designing new networks, although the current paper doesn't show much improvement. I believe there's potential to better use the introduced box filters. This paper might have *not* properly used the box filters in a conv network.

Reviewer 3



This paper present a method to compute convolutions with filters of any size with a fixed computational cost by using the integral image trick as in Viola-Jones detector [30]. This method is used to create a box convolution layer, that is a layer in which every filter is a set of boxes (one per channel) whose position and size are learnable parameters. This layer is used in a new block (Fig. 1) that is then used to replace standard dilated convolution blocks in ENet and ERFNet architectures. Due to the larger receptive field and fewer parameters of the proposed layer, the modified architectures obtain better performance than the original ones. Pros: - The idea of using integral image for convolutions and learn the filter size and position is new in my knowledge and seems to make sense when we want to read a broad proportion of the image without relying on several convutional layers. - The reported experiments seem to show that for the specific problem this box convolution makes sense and performs better than other approaches. Cons: - It seems that the authors did not mention (or not fully explain) certain important aspects of the approach such as: i) the computational cost of reading a value is 4 times more than what it would be in a normal convolution due to the integral image trick (plus the time needed to pre-compute the integrals) ii) the computational time of the full approach is then reduced because instead of using 3x3 filters, they read just average boxes and they reduce the number for channels by a factor 4 with 1x1 convolution iii) at test time, when the computational cost is actually evaluated, the authors avoid to compute the float value of the boxes (which is needed in training for learning the parameters); thus training is expected to be computationally more expensive. - I would have liked to see the idea implemented for a full comparison with a learnable convolutional filter, i.e. instead of single boxes, learn 3x3 learnable size filters and use them as a basic convolutional block. Of course, this would incur in a much higher computational cost (compared to the standard 3x3 filter), but probably also much better results. - In the experiments the places in which some blocks have been substituted by the proposed block seem a bit ad-hoc and one might consider that it is optimized to obtain the best performance gap. Additional comments: - l.179: it is not clear to me why the 1x1 convolution has a complexity of O(N^2) while the box convolution of only O(N). In my understanding the box convolution has N/4 inputs and N outputs, thus O(N^2) operations per spatial location. - The block in Fig.1 in my understanding is composed of a 1x1 convolution and the box convolution, which can be also considered 1x1 in the sense that there is a box for each channel (in my understanding). Thus, the complete block lack of spatial evaluations within channel (in the sense of 3x3 convolutions that can evaluate nearby pixels in the same channel). This might not be a problem for the given task, but it might be problematic for more difficult problems. This problem can also be compensated by the fact that the position of the box can be learned. Authors should consider to present this kind of reasoning about the approach and propose some experiments to better understand it. - The symmetry of the boxes in Fig. 2 can be due to the use of the horizontal flip of the training images for data augmentation? After rebuttal: I read the authors rebuttal and I think they managed to give an explanation to most of my concerns. In this sense I think the paper deserves publication. Still, I think it should be important to clarify that using the integral image trick makes the computation of the convolution much more expensive, but allows any size filter with the same computational cost. In this paper the proposed models are computationally cheaper because the proposed layers (based on integral image) are reduced convolutions, with spatial filters of WxHxC of 1x1x1.